# Response of essential oil hemp (*Cannabis sativa L.*) growth, biomass, and cannabinoid profiles to varying fertigation rates

**Steven L. Anderson, II**[1], **Brian Pearson**[1]*, **Roger Kjelgren**[1], **Zachary Brym**[2]

**1** Department of Environmental Horticulture, Mid-Florida Research and Education Center, Institute of Food and Agricultural Sciences, University of Florida, Apopka, Florida, United States of America, **2** Department of Agronomy, Tropical Research and Education Center, Institute of Food and Agricultural Sciences, University of Florida, Homestead, Florida, United States of America

* bpearson@ufl.edu

**Data Availability Statement:** All relevant data are within the manuscript and its Supporting Information files.

## Abstract

Five essential oil hemp (*Cannabis sativa* L.) cultivars (Cherry Blossom, Cherry Blossom (Tuan), Berry Blossom, Cherry Wine, and Cherry Blossom × Trump) were treated with six fertigation treatments to quantify the effects of synthetic fertilizer rates and irrigation electrical conductivity on plant growth, biomass accumulation, and cannabinoid profiles. Irrigation water was injected with a commercial 20-20-20 fertilizer at rates of 0, 50, 150, 300, 450, and 600 ppm nitrogen equating to 0.33 (control), 0.54, 0.96, 1.59, 2.22, and 2.85 dS m$^{-1}$, respectively. Plants were grown under artificial lighting (18 hr) to maintain vegetative growth for eight weeks, followed by an eight-week flowering period. High linear relationship between chlorophyll concentrations and SPAD-502 measurements validated the utilization of SPAD meters to rapidly identify nutrient deficiency in essential oil hemp. Cultivars expressed significant variation in plant height and cannabinoid profiles (% dry mass), in concurrence with limited biomass and cannabinoid (g per plant) yield variation. Cherry Blossom was the best performing cultivar and Cherry Wine was the least productive. Variation in plant growth, biomass, and cannabinoid concentrations were affected to a greater extent by fertilizer rates. Optimal fertilizer rates were observed at 50 ppm N, while increased fertilizer rates significantly reduced plant growth, biomass accumulation, and cannabinoid concentrations. Increased fertilizer rates (> 300 ppm N) resulted in compliant THC levels (< 0.3%), although when coupled with biomass reductions resulted in minimal cannabinoid yields. Additionally, CBD concentration demonstrated higher sensitivity to increased fertilizer rates (> 300 ppm N) compared to THC and CBG (> 450 ppm N). The results of this study can serve as a guide when using fertigation methods on essential oil hemp cultivars; although results may differ with cultivar selection, environmental conditions, and management practices.

## 1. Introduction

*Cannabis sativa* L. is a short day, herbaceous, commonly dioecious annual subshrub cultivated throughout the world for grain (seed), fiber (stems), and secondary metabolites (essential oils).

**Funding:** The funders had no role in study design, data collection and analysis, decision to publish, or preparation of the manuscript. This project was made possible by financial support from Green Roads, LLC and the UF/IFAS Office of the Dean and Research. Steven Anderson was funded by Roseville Farm's UF/IFAS Florida Industrial Hemp Endowment contribution.

**Competing interests:** The authors declare that funds were received from Green Roads, LLC, and Roseville Farms to conduct this research and fund salaries. This does not alter our adherence to PLOS ONE policies on sharing data and materials.

Following the Agriculture Improvement Act of the 2018, *Cannabis* is now categorized into two groups by the concentration of delta-9-tetrahydrocannabinol ($\Delta^9$-THC), industrial hemp ($\Delta^9$-THC < 0.3%; federally legal) and drug-type marijuana ($\Delta^9$-THC > 0.3%; federally illegal). Unlike marijuana which is exclusively cultivated for secondary metabolites (cannabinoids, terpenes, etc.), hemp cultivars have been traditionally cultivated for grain and fiber. In recent decades, high essential oil hemp cultivars have been selected for high cannabinoid secondary metabolites, led by cannabidiol (CBD) varieties; with evolving interest in varieties bred for higher levels of other cannabinoids (canabigerol, canachromine, etc) [1, 2]. Maximum essential oil production occurs within unpollinated flowers of dioecious, female *Cannabis* concentrated within glandular trichomes [3, 4]. THC and CBD are the two most abundant cannabinoids of the over 200 known phytocannabinoids [5, 6]. Their acidic precursors, tetrahydrocannabinolic acid (THCA) and cannabidiolic acid (CBDA), are synthesized from canabigerolic acid (CBGA) by THCA synthase (THCAS) and CBDA synthase (CBDAS) [7, 8], respectively. Long-standing legal restrictions of *Cannabis* growth has stunted science-based information regarding modern cultivation methods and the best management practices for optimal plant growth and physiological development of secondary metabolites.

Nutrient management is a major factor affecting plant growth and development [9, 10]. Specifically, nitrogen is the most abundant mineral nutrient in plants playing critical roles in plant development and metabolism [10]. Nitrogen supply is positively correlated to chlorophyll content in marijuana [11], although classical chlorophyll quantification can be a labor intensive, time consuming method of assessing nitrogen deficiency. SPAD chlorophyll meters are a high-throughput, noninvasive method used to grade greenness of plants [12] and potentially useful in assessing hemp nutrient deficiency. Understanding the effects of nutrient management on biomass yields in conjunction with cannabinoid production and accumulation are critically important to maintaining long-term production and economic sustainability of *Cannabis* [13]. Currently available literature lacks information pertaining to essential oil hemp fertilization rates, focusing on field grown fiber/grain hemp [14–16] and growth chamber grown marijuana cultivars [11, 17–19]. It is speculated that the recent development of high essential oil hemp cultivars were derived from the introgression of hemp haploblocks within a predominant marijuana haplotype [20, 21], indicating essential oil hemp cultivars may have similar nutrient needs as marijuana.

Recent studies have investigated the influence of fertilization rates on growth of marijuana cultivars applied through fertigation methods. When implementing fertigation management practices, over fertilization can lead to the accumulation of salts within the root zone while under fertilization leads to nutrient deficiencies and reduced growth/yield [22]. Investigation of optimal nitrogen rates during vegetative growth phases have presented varying results of 389 ppm N [17] and 160 ppm N [11] for different marijuana cultivars. Consistent with nitrogen, optimal concentration of potassium varied across marijuana cultivars [23] under vegetative growth. Fertigation rates of 389 ppm N during the vegetive growth [17] followed by 212–261 ppm N [18] during the floral period have been demonstrated to optimize biomass and cannabinoid content of select marijuana cultivars. Negative correlations have been demonstrated between cannabinoid concentrations (THC and CBG) and increased fertilizer rates [18, 24]. Empirical research defining nutrient management practices for essential oil hemp cultivars using fertigation application methods are lacking in the literature.

In this study, varying fertigation application rates were evaluated with respect to essential oil hemp cultivar production under greenhouse conditions. Five essential oil hemp cultivars were subjected to six fertigation rates throughout a complete growth cycle (vegetative and flowering) to empirically test the genetic and abiotic response of essential oil hemp to fertilizer rates. The objectives of this study were to: [i] quantify the relationship between chlorophyll

concentrations and SPAD measurements in essential oil hemp cultivars, [ii] evaluate variance in cultivar growth and cannabinoid concentrations, and [iii] quantify the effects of fertilizer rates on hemp growth parameters and cannabinoid profiles.

## 2. Materials and methods

Five dioecious essential oil hemp cultivars were sowed from seed: [i] Cherry Blossom, [ii] Cherry Blossom (Tuan), [iii] Berry Blossom, [iv] Cherry Wine, and [v] Cherry Blossom × Trump. Feminized seeds were sowed (10/14/2019; one seed per cell) within 72 round cell propagation sheets (PRO072R0G18C100) filled with Pro-Mix HP Mycorrhizae (Premier Tech Horticulture, Quakertown, PA, U.S.) media. Pro-Mix HP Mycorrhizae was used as the media in all increased pot sizes. In combination with the high porosity soil, to avoid overwatering, plants were periodically step up in pot size. Three weeks post-sowing (11/05/2019; 21 days after sowing [DAS]), seedlings were transplanted (one plant per pot) in 1.1 L square pots (SVD-450, T.O. Plastics, Clearwater, MN, U.S.) and maintained in vegetative growth for three weeks using 1000 W Metal Halide supplemental lighting (18/6 hr light/dark cycle). Plants were transplanted (11/26/2019; 43 DAS) to their final 5.68 L #2 pot (C600 Nursery Supplies, Inc., Kissimmee, FL, U.S.) and maintained in vegetative growth for two weeks to allow for root establishment. Supplemental lighting was turned off (12/11/2019; 58 DAS) and the natural daylength (~10 hr 20 min; Apopka, FL, U.S.) was used to induce vegetative-to-reproductive transition to flowering. Six weeks after floral initiation (01/21/2020), 15 cm cola (apical floral mass) samples were collected for cannabinoid analysis. Eight weeks after floral initiation (02/05/2020), plants were harvested and dried within a 70°C environment.

### 2.1 Fertilizer rates and corresponding electrical conductivity treatments

Peters Professional 20-20-20 (N-P-K) (ICL Specialty Fertilizers, Dorchester County, SC, U.S.) general purpose fertilizer with micronutrients (0.050% Mg, 0.0125% B, 0.0125% Cu, 0.050% Fe, 0.025% Mn, 0.005% Mo, and 0.025% Zn) was prepared at variable fertilizer treatment rates (0, 50, 150, 300, 450, and 600 ppm nitrogen) following their respective EC (0.33, 0.54, 0.96, 1.59, 2.22, and 2.85 dS m$^{-1}$). Electrical conductivity of irrigation supply water was constant at 0.33 dS m$^{-1}$ and thus explains the 0.33 dS m$^{-1}$ EC of the control treatment (0 ppm N). Daily fertigation (i.e., dissolved in the irrigation solution at each irrigation) treatments began upon transplanting seedling from 72 round cell propagation sheets to SVD-450 square pots (11/05/2019). Fertigation treatments were applied using Dosatron D14MZ2 injectors and stock fertilizer solutions mixed at 1:128 injection ratios of 0, 4.3, 97.4, 194.0, 291.3, and 387.9 g L$^{-1}$ fertilizer equating to 0, 50, 150, 300, 450, and 600 ppm N, respectively. Fertigation was applied with Rust MaxiJet grooved nursery pot stakes (Dundee, FL, U.S.) delivering 0.3 L min$^{-1}$ at 172.4 kPa inlet pressure. Irrigation was delivered for 4 min (1.2 L) to each SVD-450 pot followed by 20 min (6.0 L) to each of the C600 pots. The extensive leaching was utilized to maintain soil EC and avoid salt accumulation. Emitter and pour through leachate ECs were periodically monitored using the method described by Wright [25] using a Hanna instruments HI98130 (Woonsocket, RI, U.S.) pH and conductivity meter.

### 2.2 Phenotypic characterization

**2.2.1 SPAD and chlorophyll estimates.** SPAD measurements were collected using a SPAD 502 meter (Spectrum Technologies, Inc., Aurora, IL, U.S.) at the vegetative-to-floral transition to correlate to chlorophyll content for non-invasive, rapid nutrient deficiency of essential oil hemp. Chlorophyll concentrations were estimated using absorption from an Evolution 201/2020 UV-Visible Spectrophotometer (ThermoFisher Scientific, Waltham, MA, U.

S.). Chlorophyll was extracted following the modified procedure of Arnon et al. [26]. A 0.3 g sample of leaf tissue was collected from the third node below the apical meristem with fully developed leaves. Tissue samples were homogenized in 1.5 mL of 80% acetone at 15,000 rpm for 1 min. Homogenate leaf tissue was centrifuged at 10,000 rpm for 10 mins. Supernatant was diluted 1:100 (1 mL final volume) and absorption values were collected at 646, and 663 nm for chlorophyll concentration estimates [27].

**2.2.2 Plant height, growth curves, and absolute growth rates.** Plant height was measured as the distance from the media surface to the dominant apical growth point. Height measurements were collected throughout the growth cycle at 31, 36, 52, 67, 77, 99, and 114 days after sowing (DAS) to model trends in plant growth.

Temporal plant height measurements were used to fit the Weibull [28] three parameter sigmoidal function (Eq 1), where height is modeled as a function of DAS (x) with the asymptote

$$f(x) = L(1 - e^{(-(x/x_0)^b)})$$ [1]

(L), inflection point ($x_0$), and the growth rate (b) of the fitted curve. The asymptote (L [m]) is maximum value of the curve which represents maximum growth height. The inflection point ($x_0$ [d]) indicates the DAS where the slope of the logarithmic phase is at its absolute maximum. The growth rate (b) is a unitless empirical constant which defines the shape of the curve. The Weibull function was chosen over other common sigmoidal models (logistic, Gompetz, etc.) due to its flexibility to asymmetric growth allowing the inflection point to lie at any x-value. The first derivative of the Weibull function (Eq 2) estimates the absolute growth rate (AGR; m d$^{-1}$) at each time point throughout the growth cycle. Weibull parameters were estimated using the FitCurve tool in JMP (JMP®, Version 15. SAS Institute Inc., Cary, NC, 1989–2021.)

$$f'(x) = \frac{Lbe^{(-(x/x_0)^b)}\left(\frac{x}{x_0}\right)^b}{x}$$ [2]

**2.2.3 Dry biomass.** Dry biomass was measured after complete dry down within a 70°C oven. Total biomass, floral biomass, and bucked (i.e., the leaf and floral tissues) biomass was measured using an Ohaus Ranger 3000 (Parsippany, NJ, U.S.) compact bench scale. Stem diameter was measured at soil level on the date of harvest with a digital caliper.

**2.2.4 Cannabinoid analysis.** Six weeks after the vegetative to floral transition was initiated, 15 cm long apical floral samples (cola) were collected from each plant. Cola samples were cool air dried using the techniques described by Campbell and Pearson [29] for 7 days. Dried samples were ground into a fine powder using a coffee grinder and stored in 100 mL glass vials. Detailed extraction and quantification of cannabinoids methods can be found in Berthold et al. [30]. Ground samples were weighed, and cannabinoids were extracted by adding a solution of methanol and water (95:5, v/v) acidified with 0.005% formic acid at a 1:100 w/v plant material to solvent concentration ratio. Solution was vortex mixed for 5 mins, sonicated for 5 min, and centrifuged at 4°C, 3220 × g for 10 min. Supernatant was serial diluted using extraction solvent until the sample quantification fell within quantification range. Quantification of cannabinoids was conducted using a Waters I-Class Acquity UPLC (Milford, MA, U. S.) coupled with a Waters Xevo TQ-S Micro™ triple-quadrupole mass spectrometer (MS/MS) [30]. Furthermore, mass spectrometry was used for detection, no wavelengths are involved just mass transitions. The mass transitions are available in Berthold et al. [30]. Raw cannabinoid quantification and standard curve data are presented in **S1 File**.

## 2.3 Statistical inference

The experiment was conducted as a randomized complete block design (RCBD) with six replications of five cultivars (Chery Blossom, Cherry Blossom (Tuan), Berry Blossom, Cherry Wine, and Cherry Blossom × Trump) nested within six fertilizer/$EC_W$ treatments (0; 0.33, 50; 0.54, 150; 0.96, 300; 1.59, 450; 2.22, and 600 ppm N; 2.85 dS m$^{-1}$). Best linear unbiased estimator (BLUEs) of response variables ($Y_{ijk}$) were estimated in JMP (JMP®, Version 15. SAS Institute Inc., Cary, NC, 1989–2021.) by fitting Eq 3 using restricted maximum likelihood (REML) approaches with

$$Y_{ijk} = \mu + Rep(T)_{k(i)} + T_i + C_j + TC_{ij} + \varepsilon_{ijk} \qquad [3]$$

grand mean ($\mu$), fertilizer treatment effect ($T_i$), cultivar ($C_j$) effect, fertilizer treatment by cultivar interaction ($TC_{ij}$), replication nested within fertilizer treatments ($Rep(T)_{k(i)}$), and residual error ($\varepsilon_{ijk}$). The model term $Rep(T)_{k(i)}$ was fitted as a random effect, while all other terms were fitted as fixed effects. Significant statistical differences were calculated using Tukey's HSD test ($\alpha < 0.05$). Raw data (**S2 File**), variance component decomposition (**S1 Table**)**,** fixed effect tests (**S3 File**)**,** cultivar BLUEs (**S4 File**), and fertilizer rate BLUEs (**S5 File**) have been provided.

# 3. Results and discussion

## 3.1 Visual plant appearance

The visual appearance of the cultivars reflected growth responses based on genetics and fertilizer rates (**Fig 1**). Lack of fertilization (0 ppm N) resulted in restricted growth, lack of branching, and severe leaf chlorosis. The 50 ppm N and 150 ppm N treatments resulted in the tallest plants, high branching number, healthy green foliage, and lack of nutrient toxicity symptoms. Increased fertilizer rates of 300 ppm N resulted in necrosis of older, lower canopy fan leaves, growth stunting of Berry Blossom and Cherry Wine, slight leaf tip burning, and shorter branch lengths. Leaf necrosis throughout the plant architecture progressed in severity within the 450 ppm N and 600 ppm N treatments. Higher fertilizer treatments demonstrated dark green, glossy leaves with downward curling leaf architecture indicative of nitrogen toxicity. Furthermore, visual appearances indicated differential cultivar tolerances to increased fertilization rates, with Cherry Wine and Berry Blossom being the least tolerant.

## 3.2 SPAD and chlorophyll are highly correlated

SPAD measurements are a high-throughput, non-destructive measurement of chlorophyll content which can be utilized to assess nitrogen deficiencies in an array of plant species. We demonstrated high Pearson's correlations (r) between SPAD measurements and total chlorophyll (r = 0.82). Calibration curves demonstrate one SPAD unit equated to 17.4 ± 3.9 nmol mL$^{-1}$ total chlorophyll (**Fig 2**). Lack of fertilizer application (0 ppm N) resulted in the lowest SPAD reading and total chlorophyll (**S5 File**) of the fertilizer treatments. Applications of fertilizer resulted in a 1.6- (50 ppm N) to 1.8-fold (450 ppm N) increase in SPAD readings. Across fertilizer treatments (excluding control) no statistical difference were observed for total chlorophyll indicating no evidence for nitrogen deficiencies across the fertilizer rates. SPAD measurements in combination with chlorophyll estimates effectively identified a nutrient deficient (SPAD < 44) threshold for essential oil hemp. The nutrient deficient threshold was based upon the lower 95[th] percentile of the 50 ppm N treatment. Our results support the use of a SPAD meter to estimate chlorophyll content and potential nutrient deficiency quickly, nondestructively, and reliably within leaves of hemp cultivars utilized in this study.

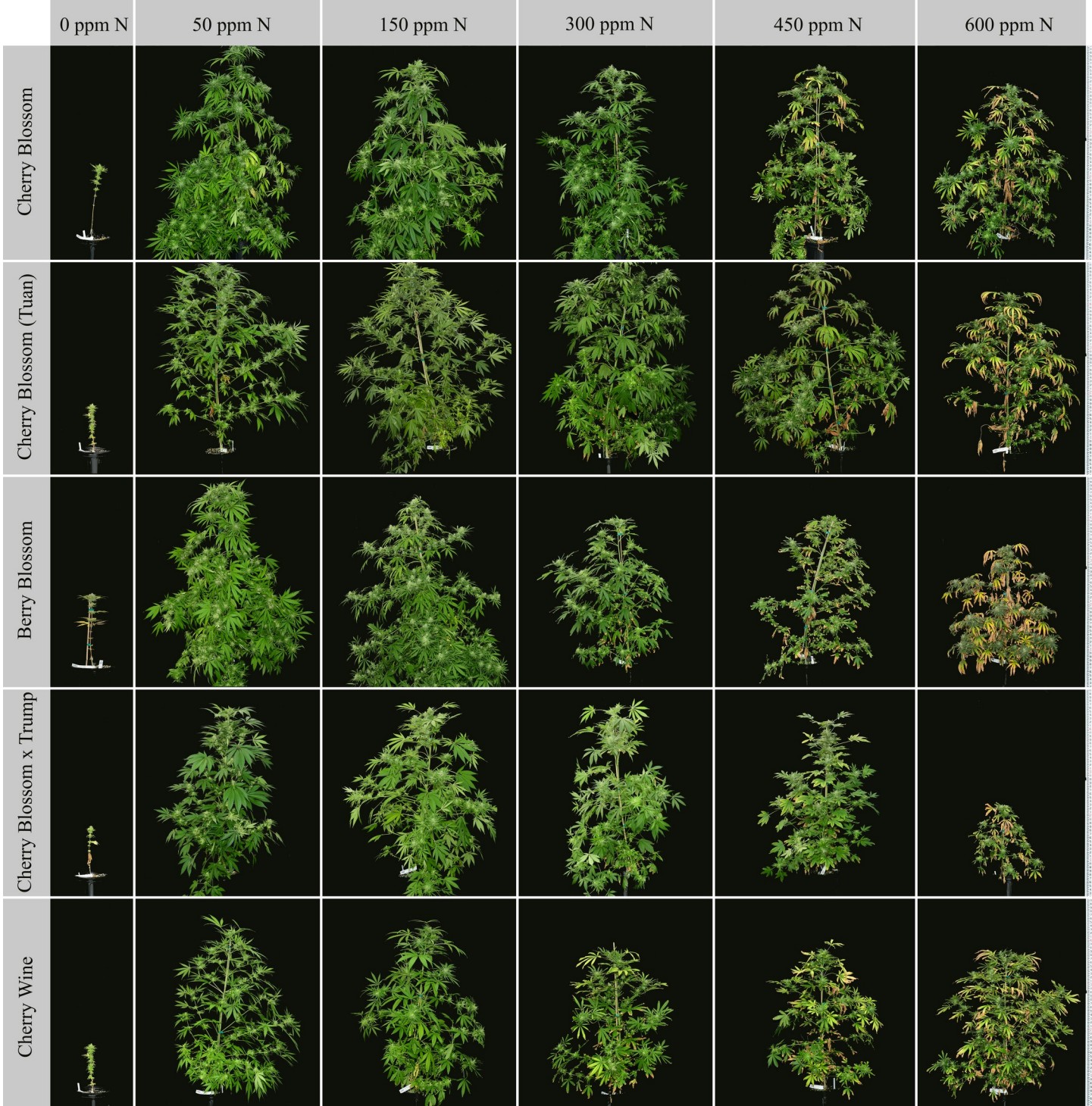

**Fig 1. Visual appearance of plants one week prior to harvest.** Columns represent fertilizer rates of 0, 50, 150, 300, 450, and 600 ppm N from left to right. Rows represent cultivars Cherry Blossom, Cherry Blossom (Tuan), Berry Blossom, Cherry Blossom × Trump, and Cherry Wine in descending order.

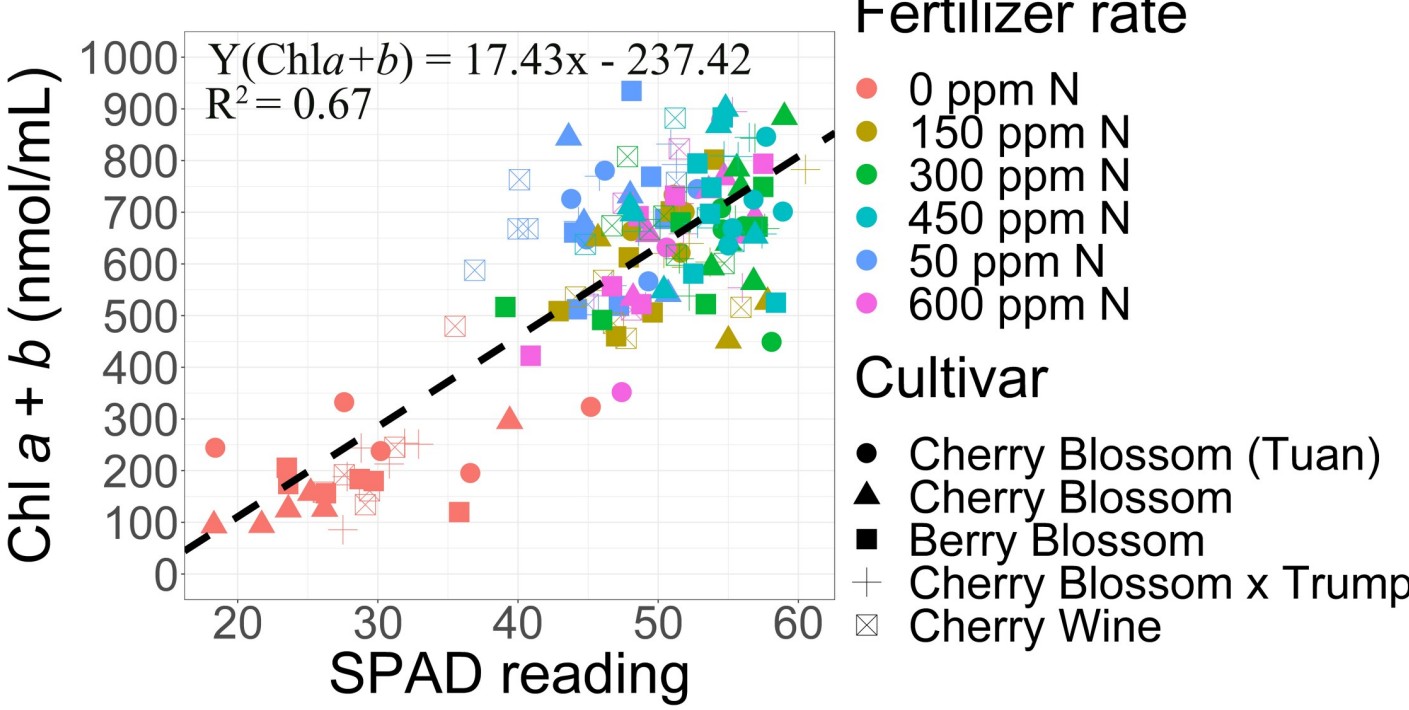

**Fig 2. Linear relationships between SPAD 502 meter and total chlorophyll (a+b) concentrations at vegetative to floral transition.**

### 3.3 Plant growth and development

**3.3.1 Genetic variation expanded following floral initiation.** Genetic variation in plant growth was observed across cultivars throughout the growth cycle with greater separation of heights at later stages of the growth cycle (**Fig 3A**). Increased growth rates were highly correlated to taller plants (r = 0.97) (**Fig 3B**). Additionally, biomass measurements were highly correlated to plant height starting at 52 DAS (r = 0.66–0.82) demonstrating that taller plants at floral initiation resulted in greater biomass yields. Unlike plant height, limited separation of biomass means were observed across cultivars (**Fig 4A**). The limited variance in cultivar biomass could be attributed to the lack of genetic diversity within essential oil hemp cultivars [31], constrained allometry [32], unstable trait expression in seeded hemp cultivars, and equivalent biomass accumulation through selection of traits beyond plant height (e.g. branch number, flower density, etc.).

**3.3.2 Increased fertilizer rates stunted plant growth.** Plant growth and development was significantly affected by fertilizer rates (**Fig 3C and 3D**). Lack of fertilizer application (0 ppm N; 0.33 dS m$^{-1}$) restricted plant development resulting in the shortest plants (**Fig 3C, red circle**), the lowest growth rates (**Fig 3D**), and limited biomass accumulation (**Fig 4A**). With the exception of the control, plant height was progressively stunted in conjunction with increased fertilizer rates resulting in the shortest plants at the highest concentration of fertilization (600 ppm; 2.85 dS m$^{-1}$). Significant reductions in plant height (compared to 50 ppm N) were observed at ≥ 600 ppm N (52 DAS), ≥ 450 ppm N (77 DAS), and ≥ 300 ppm N (99 DAS) (**S2 Table**). Such trends demonstrate reduced tolerance to elevated fertilizer rates (> 150 ppm N) as the growing season progressed. It is presumed that the increased concentrations of N supply increased the osmotic potential of leaf tissue sap, similar to marijuana cultivars, demonstrating a salinity response [11]. The gradual stunting demonstrated the time-dependent process of

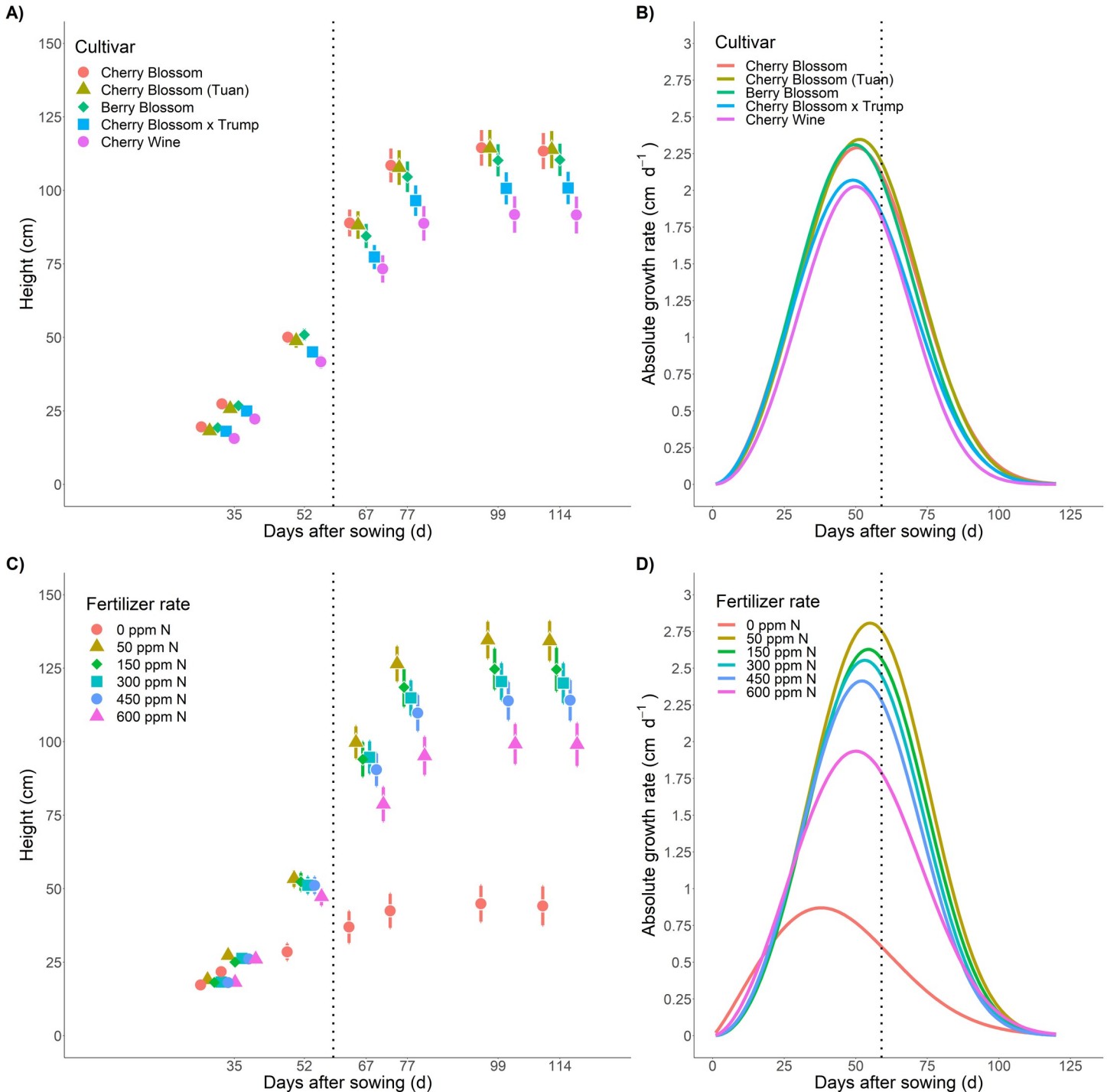

**Fig 3.** [**A**] Temporal plant heights BLUEs by cultivar. Error bars depict 95% confidence intervals. [**B**] Growth rate curves by cultivar. [**C**] Temporal plant heights BLUEs by fertilizer treatment. Error bars depict 95% confidence intervals. [**D**] Growth rate curves by fertilizer treatment. Vertical black dashed lines indicated vegetative to floral transition. Connecting letter reports can be accessed in S4 and S5 Files.

salinity response [33] with a fast response at high salinity levels early (**Fig 3C**, day 52) followed by the slow accumulation of salts within plants stunting growth at later dates (**Fig 3C**, day 99). Optimized plant growth and biomass was achieved at 50 ppm N (0.54 dS; **Fig 4B**) with

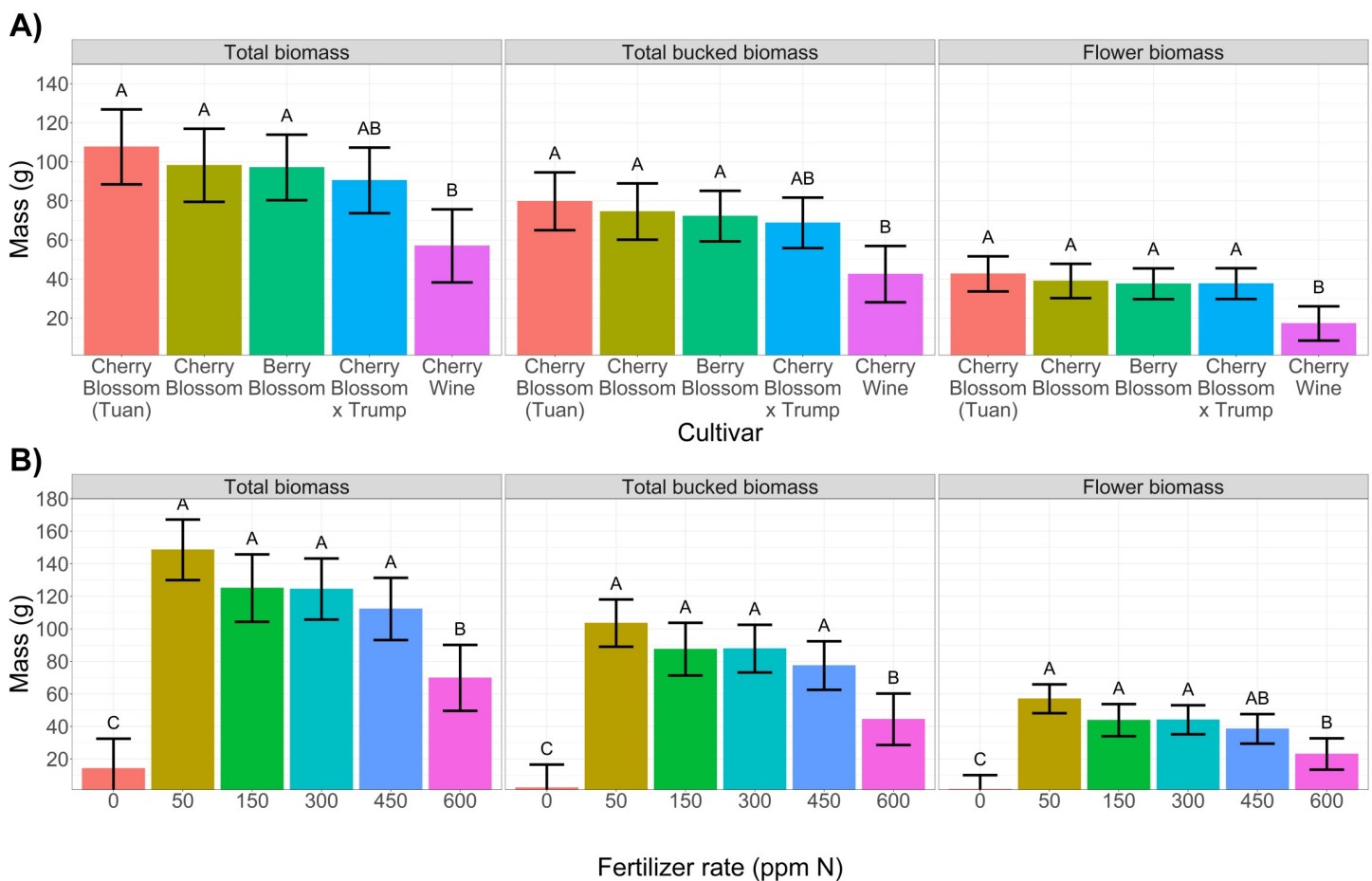

**Fig 4.** [A] Bar graphs depicting cultivar best linear unbiased estimator (BLUE) values for biomass traits total biomass, bucked biomass (flower and leaves), flower biomass (left to right). Error bars depict the 95% confidence intervals and connecting letters indicate statistical differences using Tukey's HSD (α = 0.05). [B] Bar graphs depicting fertilizer treatment BLUE values for biomass traits total biomass, bucked biomass (flower and leaves), flower biomass (left to right). Error bars depict the 95% confidence intervals and connecting letters indicate statistical differences using Tukey's HSD (α = 0.05).

statistically significant (α = 0.05) reductions in total biomass and bucked biomass at fertigation 450 ppm N (ECw > 2.22 dS) and significant reductions of floral biomass at 300 ppm N (ECw > 1.83 dS) (**Fig 4B**, **S4 File**). Although the conservative Tukey's HSD demonstrated non-significant reductions in biomass accumulation up to 450 ppm N applications, from a production mindset, implementing 50 ppm N would reduce production costs, reduce environmental impacts of fertilizer leaching, and numerically improved floral biomass by > 20% (**Fig 4B**).

To our knowledge there are no studies within the literature investigating the impacts of fertilization rates on essential oil hemp growth and biomass accumulation. Recent studies have identified the effects of fertilizer rates on growth and development of drug type marijuana cultivars. Specifically, *Cannabis* fertilization studies have focused upon the vegetative growth stages of limited cultivars grown within growth chambers/rooms. Caplan et al. [17] identified an optimal rate of 389 ppm N during vegetative growth followed by 283 ppm N during the flowering period optimized floral yields for marijuana cultivar OG Kush x Grizzly. Saloner and Bernstein [11] observed restricted growth and biomass reductions above 160 ppm N, with optimal nitrogen use efficiency between 30–80 ppm N daily fertigation of marijuana cultivar Annapurna. Differences in fertilizer formulations, nutrient holding capacity of media, size of

plants, growth rate/nutrient demand of cultivars, container size, and frequency of fertilizer applications could be attributing to the variation in optimal fertilizer rates for container grown cannabis. Furthermore, differential responses to potassium rates across marijuana cultivars [23] indicates genetic variability in the requirements and tolerances to nutrients of cannabis. Further research is needed to include greater genetic diversity and growth under common commercial production environments (greenhouse and field) focused on production methods of the *Cannabis* industry.

### 3.4 Cannabinoid accumulation

Of the 160 plants sampled, 57% exceeded the total potential THC threshold of 0.3%. Examples of cannabinoid chromatograms are available in S1 Fig. Toth et al. [34] demonstrated that only 35% of 150 $B_D/B_D$ chemotype (0.06–0.75% THC), essential oil hemp plants complied with the total potential THC threshold ($< 0.3\%$) at maturity. Our results, combined with published knowledge that cannabinoid concentrations exceed compliance thresholds around 4 weeks of flowering [35], demonstrate the need for growers to adopt temporal cannabinoid testing during the flowering stage of their crop to remain compliant with regulations.

**3.4.1 Cannabinoid variance between cultivars.** Significant variance in cannabinoid concentrations (% dry mass) were observed across cultivars (**Fig 5**). Interestingly, limited variation in cannabinoid yields (g plant$^{-1}$) existed across cultivars (**Fig 6**). Average THC concentrations at 6 weeks of flowering resulted in one cultivar (Cherry Wine) being federally compliant for harvest and sale in the U.S. As breeders continue to push the limit of THC compliance while maximizing CBD, CBG, and other secondary metabolites it will become increasingly important to understand and manage THC development and accumulation. In this study, all the varieties we tested stemmed from a common parent, Cherry Wine (https://www.leafy.com). Improved CBD concentrations and plant vigor (biomass/flower accumulation) resulted in a 3.3-fold increase in total CBD per plant (3.2 g plant$^{-1}$) of the four cultivars compared to Cherry Wine (1.0 g plant$^{-1}$) (**Figs 4A, 6, and S4 File**). The highest cannabinoid producing cultivar in this study, Cherry Blossom × Trump, is an example of a recently developed hemp cultivar possessing marijuana-based genetics. Among the cultivars cultivated in this study, Cherry Blossom × Trump likely had the greatest proportions of marijuana based genetics within its breeding pedigree [20]. Thus, the potential higher proportion of marijuana genetics may have contributed to the observation that Cherry Blossom × Trump exceeded the U.S. federal THC limit for hemp regardless of fertilizer rate when sampled at 6 weeks of flowering.

**3.4.2 Increased fertilizer rates reduced cannabinoids.** Increased fertilizer rates were negatively correlated to cannabinoid concentrations (**Fig 5**) and yield per plant (**Fig 6**). Interestingly, CBG concentrations were highest among plants that were nutrient deficient (0 ppm N; **Fig 5C**). Plants that received this same treatment (0 ppm N) also possessed greater concentrations of THC than plants that received either 450 ppm N or 600 ppm N. When coupled with total biomass accumulation, however, the applicability of no fertilizer would not be an advisable management plan. The 50 ppm N rate maximized cannabinoid concentrations (**Fig 5B**) and overall oil yields (**Fig 6B**), demonstrating optimal fertigation management maximized overall cannabinoid production. At 6 weeks of flower development, plants that received the 450 ppm N and 600 ppm N fertilizer treatments possessed mean total THC concentrations that were compliant with U.S. limits (THC $< 0.3\%$). However, these treatments also resulted in the lowest concentrations of desired cannabinoids (**Fig 5B**). Utilizing high concentrations of fertilizer to facilitate THC compliance is not advisable due to [i] reduced biomass and low cannabinoid yields, [ii] increased production costs, and [iii] increased environmental impacts due to fertilizer leaching. Our results demonstrated that lower fertilizer rates optimized

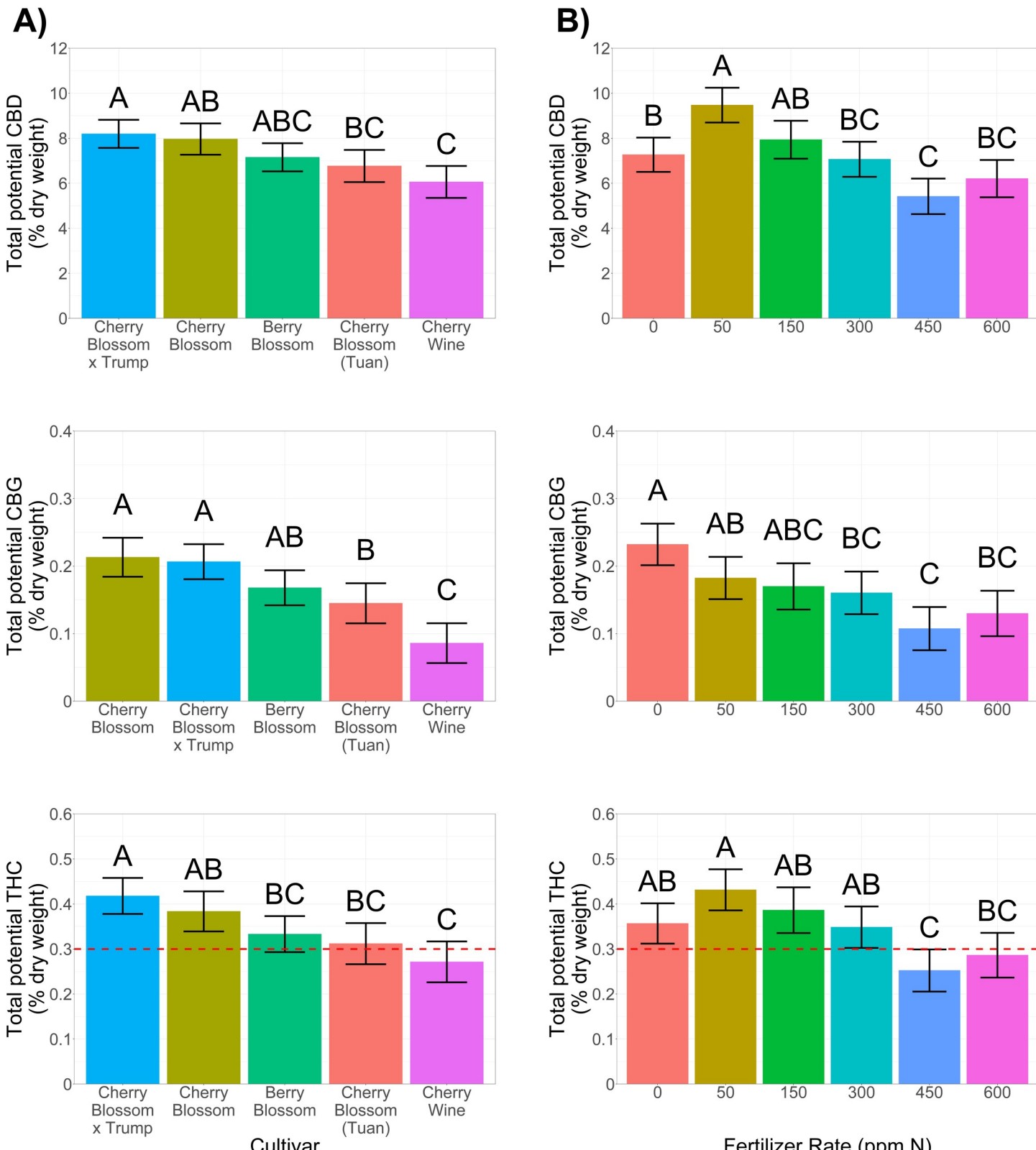

**Fig 5.** Bars represent fertilizer rate best linear unbiased estimator (BLUE) values total potential (carboxylated form + 0.877[acid form]) cannabidiol (CBD), cannabigerol (CBG), and tetrahydrocannabinol (THC) concentrations between **[A]** cultivars and **[B]** fertilizer rates. Error bars depict 95% confidence intervals and connecting letters indicate statistical differences using Tukey's HSD (α = 0.05). Horizontal red line indicates the THC compliance threshold of 0.3% total potential THC.

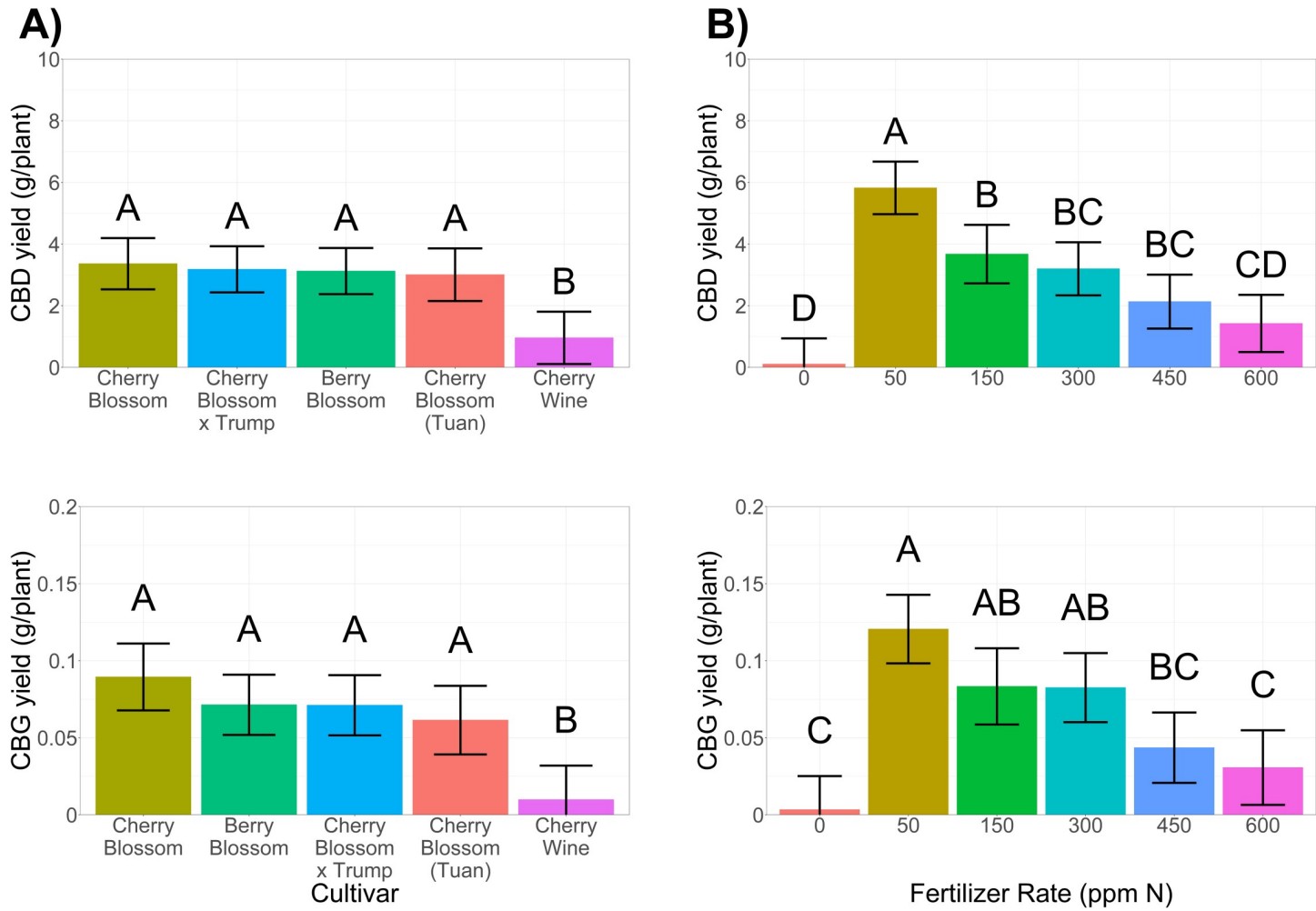

**Fig 6.** Cannabidiol (CBD; top row) and canabigerol (CBG; bottom row) across [**A**] cultivars and [**B**] fertilizer rates. Error bars depict 95% confidence intervals and connecting letters indicate statistical differences using Tukey's HSD ($\alpha = 0.05$).

cannabinoid concentrations and testing for THC compliance should be monitored throughout floral development rather than at the time of peak cannabinoid levels [35].

CBD concentrations were more sensitive to high fertilizer rates compared to other cannabinoids. More specifically, CBD concentrations were more negatively impacted than CBG or THC concentrations when plants received > 50 ppm N (**Fig 5B**). Furthermore, CBD concentrations were reduced by 16% when increasing fertilizer rated from 50 ppm N to 150 ppm N compared to 7% and 10% reductions in CBG and THC, respectively. Intriguingly, THC and CBD synthase are genetic inherited within a genomic region of low recombination [21], yet large variation in THC:CBD ratios are observed among cultivars of the same chemotype [34]. Transcriptome analysis of hemp grown under abiotic stressed environments identified up- and down-regulation genes involved in the biosynthesis of secondary metabolites in hemp [36, 37]. We speculate that abiotic stress induced by fertilization levels may have differential expression patterns of genes involved in THC and CBD biosynthesis pathways.

Fertilizer rate studies conducted in marijuana have indicated that increased fertilizer rates during the vegetative phase did not increased THC concentrations, while higher fertilizer rates resulted in reduced THC and CBG concentrations when applied during the flowering stage

[17, 18]. To our knowledge, we have presented the first empirical evidence that CBD concentrations negatively correlated to increased fertilizer rates in essential oil hemp cultivars. Furthermore, our results demonstrated that CBD concentrations and oil yield are affected by lower levels of soil EC and fertilizer rates compared to THC and CBG. Variance in cannabinoid concentrations across fertilizer rates could be due to a multitude of scenarios including: (i) differences in loci expression [38], (ii) reduced flower formation, and (iii) variation in trichome size and density [39]. Nevertheless, effective management practices coupled with reliable genetics and disciplined cannabinoid testing can result in optimized oil yields while remaining under the U.S. federal schedule I controlled substance THC threshold.

## 4. Conclusion

This research is one of the first empirical studies to report the effects of synthetic fertilizer rates and fertigation salinity levels on essential oil hemp cultivars grown in container culture. The results of this study indicated that essential oil hemp cultivars express similar irrigation salinity tolerances to vegetable crops and marijuana cultivars. This research can be used as the foundation for future essential oil hemp fertigation research to identify optimal rates and intervals between fertigation events to minimize production costs and maximize plant performance and yield. Three of the most important findings were: (i) SPAD-502 measurements were highly correlated to leaf chlorophyll content, indicating that SPAD measurements can be utilized as a rapid, non-invasive tool to access nutrient deficiency in essential oil hemp, (ii) increased fertilizer rates and irrigation salinity at maintained rates significantly reduced plant growth, biomass, and cannabinoid profiles, and, (iii) maintaining constant low rates of fertilizer available in the growing media maximized cannabinoid concentrations.

## Supporting information

**S1 Fig.** Chromatograph examples of [a] a standard curve, [b] sample N-091 below the federal THC limit of 0.3%, [c] sample N-132 near the federal THC limit of 0.3%, and [d] sample N-173 above the federal THC limit of 0.3.
(DOCX)

**S1 File. Raw cannabinoid concentration quantification data including standard curves.**
(XLSX)

**S2 File. Raw phenotype dataset.**
(CSV)

**S3 File. Fixed effects test outputs of Eq 3.**
(XLSX)

**S4 File. BLUEs estimates for cultivars from Eq 3.**
(XLSX)

**S5 File. BLUEs estimates for fertilizer rates from Eq 3.**
(XLSX)

**S1 Table. Percent variance explained by each model term of Eq 3.** Repeatability estimates calculated on an entry means basis.
(DOCX)

**S2 Table. Irrigation water salinity thresholds ($EC_W$) which caused statistically significant (Tukey's LSD; $\alpha = 0.05$) reductions in the trait of interest compared to the 50 ppm N fertilizer treatment (EC = 0.54).** Relative trait reductions indicate the reduction of the trait at or

above the $EC_W$ threshold. Percentages in parentheses indicated the percent reduction of the trait relative to the 50 ppm N fertilizer treatment (EC = 0.54).
(DOCX)

## Acknowledgments

We would like to acknowledge Brandon White, Mengzi Zhang, and Chris Halliday for their scientific and technical support; James Johnston and Dillan Raab for their hard work and effort maintaining experimental plants and collecting phenotypic data; Jerry Fankhauser and Sandra Alomar for administrative assistance; Erin Berthold and Chris McCurdy for completing the cannabinoid extractions and quantification; Green Point Research for donating the cultivars used in this research; and all members of the University of Florida IFAS Industrial Hemp Pilot Project for their collaboration.

## Author Contributions

**Conceptualization:** Steven L. Anderson, II, Brian Pearson.

**Data curation:** Steven L. Anderson, II.

**Formal analysis:** Steven L. Anderson, II.

**Funding acquisition:** Brian Pearson, Roger Kjelgren, Zachary Brym.

**Investigation:** Steven L. Anderson, II, Brian Pearson.

**Methodology:** Steven L. Anderson, II.

**Project administration:** Steven L. Anderson, II, Brian Pearson, Roger Kjelgren, Zachary Brym.

**Resources:** Steven L. Anderson, II, Brian Pearson, Roger Kjelgren, Zachary Brym.

**Supervision:** Steven L. Anderson, II, Brian Pearson, Roger Kjelgren, Zachary Brym.

**Validation:** Steven L. Anderson, II.

**Writing – original draft:** Steven L. Anderson, II.

**Writing – review & editing:** Steven L. Anderson, II, Brian Pearson, Roger Kjelgren, Zachary Brym.

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
