## [Decision Letter · Decision Letter 0]

23 Apr 2021

PONE-D-21-03958

Response of essential oil hemp (Cannabis sativa L.) growth, biomass, and cannabinoid profiles to varying fertigation rates.

PLOS ONE

Dear Dr. Anderson,

Thank you for submitting your manuscript to PLOS ONE. After careful consideration, we feel that it has merit but does not fully meet PLOS ONE’s publication criteria as it currently stands. Therefore, we invite you to submit a revised version of the manuscript that addresses the points raised during the review process.

We look forward to receiving your revised manuscript.

Kind regards,

Umakanta Sarker

Academic Editor

PLOS ONE

Journal Requirements:

"The funders had no role in study design, data collection and analysis, decision to publish, or preparation of the manuscript. This project was made possible by financial support from Green Roads, LLC and the UF/IFAS Office of the Dean and Research. Steven Anderson was funded by Roseville Farm’s UF/IFAS Florida Industrial Hemp Endowment contribution."

We note that you received funding from a commercial source: Green Roads, LLC.

3. Please remove your figures from within your manuscript file, leaving only the individual TIFF/EPS image files, uploaded separately.  These will be automatically included in the reviewers’ PDF.

Reviewers' comments:

Reviewer's Responses to Questions

**Comments to the Author**

1. Is the manuscript technically sound, and do the data support the conclusions?

Reviewer #1: Yes

Reviewer #2: Partly

Reviewer #3: Yes

2. Has the statistical analysis been performed appropriately and rigorously? 

Reviewer #1: Yes

Reviewer #2: Yes

Reviewer #3: Yes

3. Have the authors made all data underlying the findings in their manuscript fully available?

Reviewer #1: Yes

Reviewer #2: Yes

Reviewer #3: Yes

4. Is the manuscript presented in an intelligible fashion and written in standard English?

Reviewer #1: Yes

Reviewer #2: Yes

Reviewer #3: Yes

5. Review Comments to the Author

Reviewer #1: This manuscript is focused on studying impact of fertigation on plant growth, biomass accumulation, and cannabinoid profiles in 5 essential oil hemp cultivars. This study provides a baseline data of fertilizer requirements for hemp. This manuscript is written very-well and describe a technically sound piece of scientific research with data that supports the conclusions. Below are few minor comments:

1. Provide reference for L-53.

2. Mention the name of micronutrients used to prepare fertilizer treatments in L 117-118.

3. Describe how SPAD <40 threshold was obtained in L-229.

4. I would suggest to add a figure showing THC:CBD ratio for all the treatments.

Reviewer #2: This article is an interesting study reporting the effect of different N fertigation rates and CBD yielding hemp varieties on yield and cannabinoid concentrations. Some of the highlights are:

1) N fertigation rates of >150 ppm stunted plant growth demonstrating plant's salinity response.

2) The THC content for all varieties except for cherry wine was beyond 0.3%.

3) Highest cannabinoids was found in Cherry blossom*Trump because of its strong linkages to marijuana genetics

4) Nutrient related stress (low and high) possessed greater THC concentrations

Overall the paper is very well written but the conclusions are not very comprehensive. More extensive data collection and parameters need to be added to make it more conclusive. The variety selection seems to be too much narrow. The treatments need to be redesigned and reconsidered. This would be a better fit for a short communication.

Reviewer #3: Research addressing nutrient management and hemp varieties is crucial due the fact that, differently than other crops, the literature provides a scarce information for cannabis. Technically, cannabis became a new crop after some legislation changes and there is so much to be explored in this area. This manuscript is very instructive and well written. However, I found minor flaws that, if revised, it would improve the manuscript.

Line 40: This sentence could be more specific. Is CBD also sensitive at 450 and 600 ppm? Are THC and CBG sensitive at 600 ppm?

Line 100: One plant per pot? It would be good to have this information very clear.

Line 128: How the soil water content was calculated? How the 1.5 L and 6 L were calculated? Are these values the saturation point? Did the irrigation amount change as the hemp plant developed? Was there any mechanism to avoid overwatering?

6. PLOS authors have the option to publish the peer review history of their article (what does this mean?). If published, this will include your full peer review and any attached files.

Reviewer #1: No

Reviewer #2: No

Reviewer #3: **Yes: **Dr. Jose Franco Da Cunha Leme Filho

---

## [Author Response · Author response to Decision Letter 0]

12 May 2021

Reviewer #1: This manuscript is focused on studying impact of fertigation on plant growth, biomass accumulation, and cannabinoid profiles in 5 essential oil hemp cultivars. This study provides a baseline data of fertilizer requirements for hemp. This manuscript is written very-well and describe a technically sound piece of scientific research with data that supports the conclusions. Below are few minor comments:

Thank you for the kind words and your comments.

1. Provide reference for L-53.

References have been included.

2. Mention the name of micronutrients used to prepare fertilizer treatments in L 117-118.

We revised the sentence to include the micronutrient percentages. Please see lines: 117-120:

“Peters Professional 20-20-20 (N-P-K) (ICL Specialty Fertilizers, Dorchester County, SC, U.S.) general purpose fertilizer with micronutrients (0.050% Mg, 0.0125% B, 0.0125% Cu, 0.050% Fe, 0.025% Mn, 0.005% Mo, and 0.025% Zn) was prepared at variable fertilizer treatment…”

3. Describe how SPAD <40 threshold was obtained in L-229.

We included the statement below to clarify how we identified the threshold statistically. 

Line 232:“The nutrient deficient threshold was based upon the lower 95th percentile of the 50 ppm N treatment”

4. I would suggest to add a figure showing THC:CBD ratio for all the treatments.

Thank you for the suggestion. During the analysis significant differences were not discovered for CBD:THC ratios between fertilizer treatments. For this reason, we focused upon the cannabinoid concentrations as their yields are what dictate federal compliance and economic gain.

Reviewer #2: This article is an interesting study reporting the effect of different N fertigation rates and CBD yielding hemp varieties on yield and cannabinoid concentrations. Some of the highlights are:

1) N fertigation rates of >150 ppm stunted plant growth demonstrating plant's salinity response.

2) The THC content for all varieties except for cherry wine was beyond 0.3%.

3) Highest cannabinoids was found in Cherry blossom*Trump because of its strong linkages to marijuana genetics

4) Nutrient related stress (low and high) possessed greater THC concentrations

Overall the paper is very well written but the conclusions are not very comprehensive. More extensive data collection and parameters need to be added to make it more conclusive. 

Thank you for your comments. The authors believe that the results of this study are comprehensive having 7 fertilizer treatments applied to 6 hemp cultivars replicated five times within each treatment under greenhouse grown container culture. As very little published, empirical data has been produced in the cannabis research community this study creates the building blocks to justify the implementation of larger, extensive, and increasingly complicated studies. Additionally, growers are demanding science-based recommendations for this rapidly expanding new crop and delaying this information several years would not benefit the industry. We believe the results are conclusive and empirically demonstrate that maintained soil nutrient loads at higher concentrations negatively impacts the growth, biomass accumulation, and cannabinoid yield. The authors acknowledge that additional studies should be conducted to fine tune the exact nutrient application rate and timing within container and field grown conditions, but studies such as ours are essential prior to investing resources to answer such questions.

The variety selection seems to be too much narrow. 

The genetic diversity of hemp is quite narrow and the lack of genetic stability accompanying a cultivars name makes it increasingly difficult to evaluate genetic diversity on a cultivar name basis. These cultivars were donated to our research efforts and we made the best use of them. Due to limited funding of hemp research, we could not justify DNA sequencing to evaluate the diversity of these cultivars. The closest related study included only 2 marijuana cultivars which we have expanded the genetic comparison within the allotted greenhouse space. We have provided insight and data to justify future research efforts with greater numbers of cultivars.

The treatments need to be redesigned and reconsidered. This would be a better fit for a short communication.

The authors are not sure what the reviewer means by “the treatments need to be redesigned and reconsidered”. Without providing an explanation it is difficult to improve the manuscript with such comments. This research provides critical empirical data which can be expanded upon in future research projects which the authors alluded to in the discussions of results.

Reviewer #3: Research addressing nutrient management and hemp varieties is crucial due the fact that, differently than other crops, the literature provides a scarce information for cannabis. Technically, cannabis became a new crop after some legislation changes and there is so much to be explored in this area. This manuscript is very instructive and well written. However, I found minor flaws that, if revised, it would improve the manuscript.

Thank you for your comments!

Line 40: This sentence could be more specific. Is CBD also sensitive at 450 and 600 ppm? Are THC and CBG sensitive at 600 ppm?

Great question. The sensitivity is based on statistically significant reductions in the cannabinoids which continues to be significantly reduced at higher concentrations then presents. We have clarified this with the inclusion of greater than symbols.

“increased fertilizer rates (>300 ppm N) compared to THC and CBG (>450 ppm N).”

Line 100: One plant per pot? It would be good to have this information very clear.

Clarified:

“Feminized seeds were sowed (10/14/2019; one seed per cell) within 72 round cell propagation sheets (PRO072R0G18C100) filled with Pro-Mix HP Mycorrhizae (Premier Tech Horticulture, Quakertown, PA, U.S.) media. Three weeks post-sowing (11/05/2019; 21 days after sowing [DAS]), seedlings were transplanted (one plant per pot) in 1.1 L square pots (SVD-450, T.O. Plastics, Clearwater, MN, U.S.)”

Line 128: How the soil water content was calculated?

To maintain the consistent soil EC we choose to extensively leached each day to avoid EC build up, but exact soil water content was not considered here.

How the 1.5 L and 6 L were calculated?

These are the volume of irrigation water the emitters displaced during the irrigation events. We have corrected a typo:

L131: Fertigation was applied with Rust MaxiJet grooved nursery pot stakes (Dundee, FL, U.S.) delivering 0.3 L min-1 at 172.4 kPa inlet pressure. Irrigation was delivered for 4 min (1.2 L) to each SVD-450 pot followed by 20 min (6.0 L) to each of the C600 pots.

 Are these values the saturation point? Did the irrigation amount change as the hemp plant developed? Was there any mechanism to avoid overwatering?

Great question. Over watering is very evident in cannabis as the plants will wilt and die overnight. By stepping up the pot size during the experiment and using high porosity soil we were able to avoid such conditions which would have been evident in the control planting. 

Additionally, we acknowledge that future research is necessary to track irrigation necessity and proper fertilizer application timing. Our findings provide insight as to optimal maintained nutrient levels for essential oil hemp growth.

This was clarified in line 104:

“In combination with the high porosity soil, to avoid overwatering, plants were periodically step up in pot size.”

Reviewer #4: I found the article presents laudable data on hemp a widely used bioresource. The information presented will be of benefit to readers. The results are of some scientific importance for cultivation of essential oil cultivars.

Scientific composition and writing are fine, and the manuscript can be accepted after some trivial queries.

• Please revise the English.

The author’s native langue is English. Without specific details we believe the English is well written and appropriate. We have done our best to review and correct any obvious writing errors.

• Do not justify your text in the whole manuscript. Please left align only.

Within the PLOS ONE manuscript body formatting guidelines the text is justified. For this reason, we will leave our text body justified.

• Write the title as sentences format. (check the format of journal)

Corrected.

• Write the first line of each part without hanging. (check the format of journal)

Paragraphs are indented within the PLOS ONE manuscript body formatting guidelines.

Introduction: 

1. Is it possible to give some explanations based on the literature on the role of fertilizer in improving growth and chlorophyll?

2. Is it possible to give some explanations about the necessity of measurement of chlorophyll and SPAD? 

The authors feel the above 2 comments were adequately addressed within the introduction.

Line: 64: Nutrient management is a major factor affecting plant growth and development [9, 10]. Specifically, nitrogen is the most abundant mineral nutrient in plants playing critical roles in plant development and metabolism [10]. Nitrogen supply is positively correlated to chlorophyll content in marijuana [11], although classical chlorophyll quantification can be a labor intensive, time consuming method of assessing nitrogen deficiency. SPAD chlorophyll meters are a high-throughput, noninvasive method used to grade greenness of plants [12] and potentially useful in assessing hemp nutrient deficiency.

Materials and methods:

1. L100 please explain about these cultivars like: "Essential oil cultivars are bred and grown for their essential oils (cannabinoids and terpenes)".

This information was addressed in the introduction and the authors feel it does not need such clarification in the methods.

Line 53: “In recent decades, high essential oil hemp cultivars have been selected for high cannabinoid secondary metabolites, led by cannabidiol (CBD) varieties; with evolving interest in varieties bred for higher levels of other cannabinoids (canabigerol, canachromine, etc).”

Also, explain whether the essential oil cultivars are dioecious or monoecious? 

They can be either or, but are commonly dioecious. It is thought that the monecious phenotype is caused by recessive genes that are fixed during excessive inbreeding.”

Clarified: 

Line 100: Five dioecious essential oil hemp cultivars were sowed from seed.

2. L102 Feminized seeds were sowed. It is better to omit Feminized.

The authors believe this is a necessary detail to include as to avoid the discussion of rouging males within the text.

3. L105 what culture medium were the square pots filled with?

Clarified:

Line 104: Pro-Mix HP Mycorrhizae was used as the media in all increased pot sizes.

4. In heading of section 2.1. Fertilizer rates and corresponding electrical conductivity treatments

 L118 fertilizer with micronutrients…Please insert the analysis of fertilizer.

The authors are not sure what the reviewer is requesting as “analysis of fertilizer”. We have included the percentages of micronutrients as requested by another reviewer.

Line 120: Peters Professional 20-20-20 (N-P-K) (ICL Specialty Fertilizers, Dorchester County, SC, U.S.) general purpose fertilizer with micronutrients (0.050% Mg, 0.0125% B, 0.0125% Cu, 0.050% Fe, 0.025% Mn, 0.005% Mo, and 0.025% Zn)

5. In heading of section 2.2.1 SPAD and chlorophyll sstimates correct to Estimates

Thank you. Corrected.

6. In part 2.2.2 Plant height, growth curves, and absolute growth rates:

 L145 It is better to use crown instead of the media surface.

Thank you for the comment. As we have already completed the experiment, we will keep this in mind for future research.

 Mentioning the references of Eq1 and Eq2 are essential. 

Eq. 2 is the derivative of Eq.1 as a result we have cited the appropriate literature. Weibull, 1951 was added as a reference.

7. In part 2.2.4 Cannabinoid analysis:

 please insert wavelength absorbance of samples and standard curve for mass spectrometer.

Thank you for the comment.. If readers would like to learn more about the process they can reference Berthold et al. 2020 from within the text. In general mass spectrometry was used for detection so no wavelengths are involved just mass transitions. Wavelengths would be relevant if using PDA or QDA detection or some type of UV detection. The mass transitions are available in my Berthold et al. 2020.

We included a statement on L186:

“Furthermore, mass spectrometry was used for detection, no wavelengths are involved just mass transitions. The mass transitions are available in Berthold et al. 2020. Raw cannabinoid quantification and standard curve data are presented in S1 File.”

Results

I read the results and discussion; the flow is OK.

1. At least one chromatogram of cannabinoids must be presented in the text.

The authors believe this would take away from the manuscript as it is not focused upon methodology of cannabinoid quantification. We have provided some chromatogram figures within the supplemental files.

L314: Examples of cannabinoid chromatograms are available in S1 Fig.

2. Please insert the table of analysis of variance in MS.

Thank you for the comment. We conducted our research using mixed linear models and REML approaches. Mean squares are not appropriate for such an approach and were not presented. We have included the variance component decomposition within the S1 Table and the fixed effect tests in S3 File. 

We have included a statement to inform readers of the data:

L204: “Significant statistical differences were calculated using Tukey’s HSD test (α < 0.05). Raw data (S2 File), variance component decomposition (S1 Table), fixed effect tests (S3 File), cultivar BLUEs (S4 File), and fertilizer rate BLUEs (S5 File) have been provided.”

3. Fig 3. (a and c) are not clear…also, it should be inserted the alphabet to show the significant differences between treatments.

Thank you for the comment. The figures are demonstrating the grouping BLUEs (i.e., the points) and the 95% confidence intervals. Overlapping confidence intervals indicate non-significant differences between groups. Adding the connecting letters would clutter the figure and likely be un-readable. 

We have indicated within the figure caption that readers interested in the connecting letters can reference S4 File and S5 File.

Reference

Check referencing format well. (The name of the journal should be abbreviated.)

We have checked the reference formatting.

---

## [Decision Letter · Decision Letter 1]

20 May 2021

PONE-D-21-03958R1

Response of essential oil hemp (Cannabis sativa L.) growth, biomass, and cannabinoid profiles to varying fertigation rates.

PLOS ONE

Dear Dr. Anderson,

Thank you for submitting your manuscript to PLOS ONE. After careful consideration, we feel that it has merit but does not fully meet PLOS ONE’s publication criteria as it currently stands. Therefore, we invite you to submit a revised version of the manuscript that addresses the points raised during the review process.

ACADEMIC EDITOR:

Check carefully the whole MS, missing of spacing (many), absence of spacing in different symbols such as “>” “<” “=” “≥” including figure captions and figures.

Line 63: Include “the” before the word “best”

Line 183: ---at 4°C, 3220 ×g---. Use space after “4” and after “×”

Line 194: ---fertilizer/ ECW---. Delete the space after “/”. Change small “x” to the cross symbol “×”. Follow this style for whole MS e. g., line 275, 307, 309,

Figures 4-6: Check the error bars. It seems too high. Also, seems that the lettering doesn’t in accordance with error bars.

We look forward to receiving your revised manuscript.

Kind regards,

Umakanta Sarker

Academic Editor

PLOS ONE

Journal Requirements:

Reviewers' comments:

Reviewer's Responses to Questions

**Comments to the Author**

1. If the authors have adequately addressed your comments raised in a previous round of review and you feel that this manuscript is now acceptable for publication, you may indicate that here to bypass the “Comments to the Author” section, enter your conflict of interest statement in the “Confidential to Editor” section, and submit your "Accept" recommendation.

Reviewer #3: All comments have been addressed

Reviewer #4: All comments have been addressed

2. Is the manuscript technically sound, and do the data support the conclusions?

Reviewer #3: Yes

Reviewer #4: Yes

3. Has the statistical analysis been performed appropriately and rigorously? 

Reviewer #3: Yes

Reviewer #4: Yes

4. Have the authors made all data underlying the findings in their manuscript fully available?

Reviewer #3: Yes

Reviewer #4: Yes

5. Is the manuscript presented in an intelligible fashion and written in standard English?

Reviewer #3: Yes

Reviewer #4: Yes

6. Review Comments to the Author

Reviewer #3: Great work! Very important scientific information can be release with the publication of this manuscript.

Reviewer #4: The authors have adequately addressed the comments raised in a previous round of review and I feel that this manuscript is now acceptable for publication.

7. PLOS authors have the option to publish the peer review history of their article (what does this mean?). If published, this will include your full peer review and any attached files.

Reviewer #3: **Yes: **Jose Franco Da Cunha Leme Filho

Reviewer #4: **Yes: **Mahnaz Abdollahi

---

## [Author Response · Author response to Decision Letter 1]

24 May 2021

Check carefully the whole MS, missing of spacing (many), absence of spacing in different symbols such as “>” “<” “=” “≥” including figure captions and figures.

That you for the comment. We have reviewed the manuscript and made the appropriate corrections.

Line 63: Include “the” before the word “best”

Corrected.

Line 183: ---at 4°C, 3220 ×g---. Use space after “4” and after “×”

Corrected.

Line 194: ---fertilizer/ ECW---. Delete the space after “/”. Change small “x” to the cross symbol “×”. Follow this style for whole MS e. g., line 275, 307, 309,

Corrected.

Figures 4-6: Check the error bars. It seems too high. Also, seems that the lettering doesn’t in accordance with error bars.

Thank you for the comment we have double check the figures and the error bars are correct, indicating the 95% confidence intervals. The connecting letters were also checked, show no sign of differing from the error bars, and are consistent with the original JMP output.

---

## [Editor Report · Decision Letter 2]

27 May 2021

Response of essential oil hemp (Cannabis sativa L.) growth, biomass, and cannabinoid profiles to varying fertigation rates.

PONE-D-21-03958R2

Dear Dr. Anderson,

We’re pleased to inform you that your manuscript has been judged scientifically suitable for publication and will be formally accepted for publication once it meets all outstanding technical requirements.

Kind regards,

Umakanta Sarker

Academic Editor

PLOS ONE
---

## [Editor Report · Acceptance letter]

9 Jun 2021

PONE-D-21-03958R2 

Response of essential oil hemp (Cannabis sativa L.) growth, biomass, and cannabinoid profiles to varying fertigation rates 

Dear Dr. Anderson II:

I'm pleased to inform you that your manuscript has been deemed suitable for publication in PLOS ONE. Congratulations! Your manuscript is now with our production department. 

Kind regards, 

on behalf of

Professor Umakanta Sarker 

Academic Editor

PLOS ONE